# OPEN-WORLD TEST-TIME TRAINING:
# SELF-TRAINING WITH CONTRASTIVE LEARNING

## ABSTRACT

Traditional test-time training (TTT) methods, while addressing domain shifts, often assume a consistent class set that limits their applicability in real-world scenarios with infinite variety. Open-World Test-Time Training (OWTTT) addresses the challenge of generalizing deep learning models to unknown target domain distributions, especially in the presence of strong Out-of-Distribution (OOD) data. Existing TTT methods often struggle to maintain performance when confronted with strong OOD data. In OWTTT, the primary focus has been on distinguishing between strong and weak OOD data. However, during the early stages of TTT, initial feature extraction is hampered by interference from strong OOD and corruptions, leading to reduced contrast and premature classification of certain classes as strong OOD. To handle this problem, we introduce Open World Dynamic Contrastive Learning (OWDCL), an innovative approach that leverages contrastive learning to augment positive sample pairs. This strategy not only enhances contrast in the early stages but also significantly enhances model robustness in later stages. In comparison datasets, our OWDCL model achieves state-of-the-art performance.

## 1 INTRODUCTION

Deep neural networks (DNNs) have demonstrated remarkable performances across many application scenarios with well-prepared datasets Amodei et al. (2016); He et al. (2016); Liu et al. (2021d). These successes typically rely on the assumption of independent and identically distributed (i.i.d.) data, meaning that training and test data are drawn from the same distribution. However, in real-world settings, meeting this requirement is impractical Mirza et al. (2023). For instance, applying the assumption to self-driving tasks may fail due to unpredictable elements like fog, snow, rain, rare traffic incidents, or unusual obstacles like sandstorms and characters in strange costumes. In medical diagnosis, the variance in equipment noise and diverse physiological characteristics of patients may compromise the model's efficacy.

In real-world scenarios, the i.i.d. assumption often breaks down due to variable noise from different device sensors, as well as weather and climate conditions. This leads to a domain shift between the training and test sets, resulting in models that perform well on training data but fail on real-world test data Hendrycks & Dietterich (2019). Addressing this discrepancy is essential for developing robust models that can effectively handle real-world variability. In practical scenarios, target domain data is often unavailable until inference, necessitating immediate, reliable test data predictions without extra interventions. This is vital in time-sensitive or resource-limited settings where rapid adaptation is key. Test-time training/adaptation (TTT/TTA) tackles this by rapidly reducing domain shift and boosting model performance, using unlabeled target domain data during inference Liu et al. (2021c); Wang et al. (2020); Sun et al. (2020). Recent TTT advancements show promise, employing meta-learning Bartler et al. (2022) for swift task adaptation, student-teacher frameworks Sinha et al. (2023) for knowledge distillation under domain shift, and adversarial sample techniques Croce et al. (2022) for enhanced robustness and adaptability.

TTT methods, which rely on unlabeled target domain data to address domain shifts during testing, may struggle with varying levels of strong OOD data. Recent advancements in OWTTT tackle this issue by dynamically expanding prototypes based on the feature distribution of the source domain, thereby improving the distinction between weak and strong OOD data Li et al. (2023). However, a

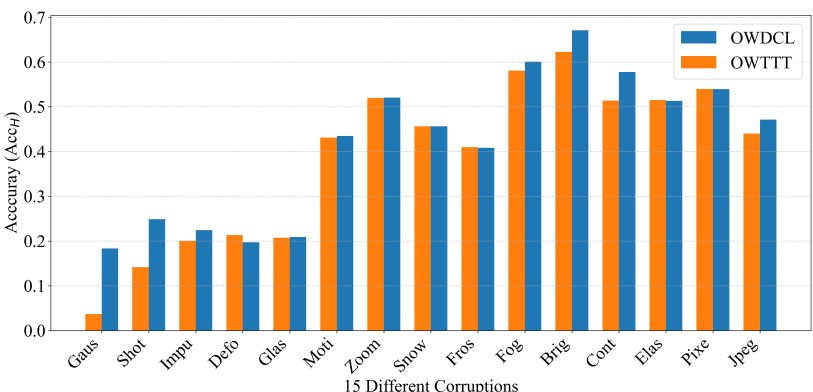

Figure 1: In an experimental setup involving 15 types of corruption within the ImageNet-C dataset and employing the MNIST dataset as a benchmark for Strong OOD analysis, we conduct a performance comparison between OWDCL and OWTTT.

key prerequisite for these methods is the model's ability to initially extract features from weak OOD data. Without this capability, weak OOD data—potentially indistinguishable from strong OOD under significant domain shifts—may be mistakenly treated as noise, leading to its misclassification as strong OOD during the TTT phase. In this paper, we address the challenge of initial domain shifts during testing, where the model encounters a scarcity of positive samples, often resulting in the misclassification of weak OOD data as strong OOD noise. Inspired by contrastive learning Chuang et al. (2020), we propose that augmented samples should maintain the same feature distribution as their originals. To tackle the challenges of the early TTT stage, where samples lacking contrast can be indistinguishable from strong OOD, our approach employs simple data augmentation to generate positive sample pairs (see in Figure 1). We incorporate the NT-XENT contrastive learning Chen et al. (2020) loss function, utilizing these pairs to assist the model's adaptation and prevent premature classification of classes as strong OOD due to initial feature extraction difficulties. Subsequently, we align these pairs with the source domain class cluster centers, enhancing the robustness of our method and enabling basic clustering for strong OODs. We term this methodology Open World Dynamic Contrastive Learning (OWDCL).

The contributions of this paper can summarized as follows:

- We propose a novel self-training with contrastive learning for open-world test-time training (OWDCL). Notable, OWDCL introduces no extra network modules over the backbone network, making it simple to implement and computationally efficient.

- Our approach is the first work to introduce contrastive learning as a method for reducing domain shifts in open-world test-time training (OWTTT) problems.

- Extensive experiments on several open-world benchmarks, including CIFAR10/CIFAR100 and ImageNet demonstrate that OWDCL can consistently yield significant performance improvements.

## 2 METHODS

### 2.1 PROBLEM FORMULATION

Test-time training aims to adapt the source domain pre-trained model to the target domain which may be subject to a distribution shift from the source domain. So we define the source domain data as $\mathcal{X}_s$, and target domain data as $\mathcal{X}_t$. we also define the source label as $Y_s = \{1, 2, ..., m\}$, the strong OOD label set as $Y_{str} = \{m+1, ..., m+n\}$, and the target label as $Y_t = Y_s \cup Y_{str}$. To clarify, we define **weak Out-of-Distribution (weak OOD)** as those classes that align with the source domain yet are subjected to alterations like noise or other forms of corruption. In contrast, **strong Out-of-**

**Distribution (strong OOD)** encompasses categories that are entirely new and distinct from those of the source domain. The overall framwork of the proposed method is illustrated in Figure 2.

Before the TTT stage, We will extract the features of the source domain $\mathcal{X}_s$ through the pre-training model $\mathcal{F}(\cdot)$, and summarize the distribution of the source domain label features $\mathcal{D}_s = \{d_1^s, ..., d_m^s\}$. At the official start of the TTT stage, We augment the sample $x_i$ by data augmentation to obtain the positive sample pair $x_i'$, they have the same label $y_i \in Y_t$. According to the threshold $\tau$, the label of $x_i$ is determined through $\mathcal{D}_s$ and the comprehensive between $x_i$ and $x_i'$. If it is not in $\mathcal{D}_s$, it is divided into $\mathcal{D}_{str} = \{d_{m+1}^{str}, ...., d_{m+n}^{str}\}$. Since there is no label in open-world TTT, we will set a pseudo-label $\hat{y}_i \in Y_t$ based on sample $x_i$.

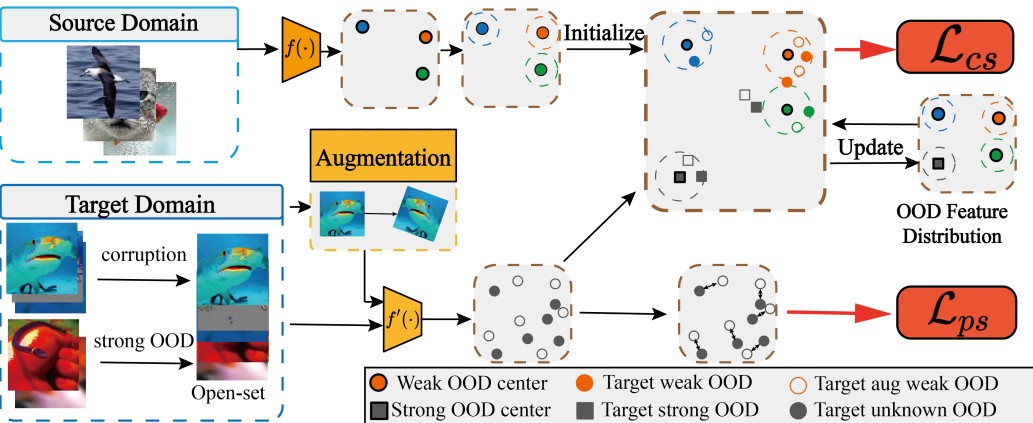

Figure 2: Overall framework of our model OWDCL. (1) $\mathcal{L}_{ps}$: Improve the feature extraction ability of the model by comparing samples with enhanced samples. (2)$\mathcal{L}_{cs}$: The classification accuracy is optimized through the comprehensive comparison between the enhanced sample pair and the class center of gravity.

## 2.2 OVERALL TEST-TIME TRAINING FRAMEWORK

In comparison with Test-Time Adaptation, Test-Time Training allows for the use of a subset of source domain data. However, due to the requirement for low latency, it does not permit access to the entire source domain dataset. Given this constraint and the demonstrated effectiveness of cluster structures in domain adaptation tasks Saito et al. (2018), their application is maintained in open-world TTT Li et al. (2023). Feature extraction from the source domain $\mathcal{X}_s$ will be performed using the pre-trained model $\mathcal{F}(\cdot)$. The cluster centers for each class are defined as follows:

$$d_m = \frac{1}{M} \sum_{i=1}^{M} \mathcal{F}(x_i), y_i \in Y_S \tag{1}$$

where $M$ represents the number of samples for a class in the source domain.

Existing research Li et al. (2023) show excellent performance in most scenarios for open-world test-time training. However, in certain cases, while the discrimination of strong OOD instances improves, there is a noticeable decline in handling weak OOD instances, as illustrated in Figure 1. At the onset of TTT, some classes are ineffectively classified, with accuracy deteriorating as TTT progresses. This is common in TTT/TTA, where models, lacking target domain labels and facing corruption interference, often use entropy-like methods to minimize output confusion Wang et al. (2020); Niu et al. (2022). Ineffective initial feature extraction of specific classes leads to misclassification as noise. This challenge is exacerbated in open-world TTT, compounded by corruption and strong OOD disturbances, making the unsupervised process more complex.

Current research often neglects to enhance feature extraction capabilities for individual samples, instead focusing on distinguishing between strong and weak OOD scenarios. We believe this issue arises in the early stages of the model, where the lack of labels and class corruption hampers effective feature extraction, leading to insufficient comparisons and feedback. Inspired by contrastive

learning He et al. (2020); Chen et al. (2020); Chen & He (2021), we utilize simple data augmentation techniques to enhance input samples. More complex augmentations, such as adjustments to contrast and brightness combined with corrupted data, can hinder model convergence. Therefore, for $x_i$, we employ flipping and a random rotation ranging from 0 to 30%, resulting in augmented data $x_i'$. Regarding the data enhancement strategy, we opt for simple rather than novel or complex data augmentations to facilitate comparative learning with sample pairs. Our experiments demonstrate that several sets of basic data enhancements yield similar effects. Specifically, a combination of vertical flipping and rotation within 0-15/45 degrees appears to be most effective. This approach is chosen for its simplicity and effectiveness. It is important to note that we advise against using contrast adjustments and adding other forms of noise for data enhancement. This is because weak OOD samples may already exhibit such corruptions, and complex augmentations could lead to convergence difficulties during testing.

Based on the previous analysis, for the samples $x_i$ and their augmented counterparts $x_i'$, the model $\mathcal{F}(\cdot)$, as derived from pre-training, and its iteratively updated version during the Test-Time Training (TTT) process, $\mathcal{F}'(\cdot)$, are believed to adhere to the following mathematical relation:

$$\mathcal{F}'(x_i) = \mathcal{F}'(x_i') \tag{2}$$

Based on this hypothesis, we implement contrastive alignment using positive sample pairs as well as contrastive alignment through clusters and sample pairs.

### 2.3 Contrastive Alignment by Positive Sample Pairs

For each sample $x_i$ and its augmented counterpart $x_i'$ in the current batch, we extract features $\mathcal{F}'(x_i)$ and $\mathcal{F}'(x_i')$ using the model $\mathcal{F}'(\cdot)$. The first step involves normalizing these features with the L2 norm, calculated as:

$$\|\mathbf{v}\|_2 = \sqrt{v_1^2 + v_2^2 + \ldots + v_n^2} \tag{3}$$

Then he result post-normalization using the L2 norm is articulated as:

$$v_i = \frac{\mathcal{F}(x_i)}{\sqrt{\sum_{i=1}^{B} \mathcal{F}'(x_i)^2}},$$
$$v_i' = \frac{\mathcal{F}(x_i')}{\sqrt{\sum_{i=1}^{B} \mathcal{F}'(x_i')^2}} \tag{4}$$

where $B$ is the number of samples in the current batch.

Based on Eq. 4, we then compute the similarity among pairs of positive samples within the normalized vectors as follows:

$$\mathcal{S}(v_i, v_j')_{pos} = \exp(\frac{\sum_{i,j=1}^{B} v_i \cdot v_j'}{\gamma_1}) \tag{5}$$

where $\gamma_1$ represents the temperature normalization factor, which scales the outcome.

Subsequently, the similarity among pairs of negative samples is computed using a different formula, as outlined below:

$$\mathcal{S}(v_i, v_j')_{neg} = \exp(\frac{v_i) \cdot v_j'^T}{\gamma_1}),$$
$$\mathcal{S}(v_i', v_j)_{neg} = \exp(\frac{v_i' \cdot v_j^T}{\gamma_1}) \tag{6}$$

In conclusion, by leveraging the identified similarities and differences in both positive and negative sample pairs, we utilize the Normalized Temperature-Scaled Cross-Entropy Loss (NT-XENT) Chen et al. (2020) for optimization. This loss function excels at discerning relational dynamics between

data points in the absence of labeled data, while avoiding comparisons between identical samples. The final loss formulation for the initial phase is expressed as:

$$
\begin{aligned}
\mathcal{L}_{ps} = & - \alpha_1 (\log(\frac{\mathcal{S}(v_i, v'_j)_{pos}}{\sum_{k \neq i}^{B} \mathcal{S}(v'_i, v_k)_{neg} + \mathcal{S}(v_i, v'_j)_{pos}}) \\
& + \log(\frac{\mathcal{S}(v_i, v'_j)_{pos}}{\sum_{k \neq j}^{B} \mathcal{S}(v'_k, v_j)_{neg} + \mathcal{S}(v_i, v'_j)_{pos}}))
\end{aligned}
\tag{7}
$$

where $\alpha_1$ is a hyper-parameter that adjusts the impact magnitude of the loss.

Optimizing the $\mathcal{L}_{ps}$ loss function enables the model to defer classifying a class as strong OOD until it has effectively extracted features from that class's samples. This approach enhances the efficacy of each sample within the weak OOD class, ensuring more precise and discriminative feature extraction.

## 2.4 CONTRASTIVE ALIGNMENT BY CLUSTER AND SAMPLE PAIRS

For each sample $x_i$, the strong OOD score is quantified based on its degree of similarity to the nearest cluster center $d_k$ in the source domain. $< \cdot, \cdot >$ measures the cosine similarity. This quantification is defined as follows:

$$
os_i = 1 - \max_{d_k \in \mathcal{D}_s} \langle \mathcal{F}'(x_i), d_k \rangle
\tag{8}
$$

Building on insights from previous research, we establish the optimal threshold as the boundary that distinguishes between two distinct distribution patterns. This approach conceptualizes the classification of outliers into two separate clusters, which can be defined as follows:

$$
\begin{aligned}
N^+ = \sum^i \mathbb{1}(os_i > \tau), \\
N^- = \sum^i \mathbb{1}(os_i \leq \tau)
\end{aligned}
\tag{9}
$$

where $\mathbb{1}(\cdot)$ is the indicator function. The optimal threshold $\tau^*$ is identified by optimizing:

$$
\min_{\tau} \frac{1}{N^+} \sum_i [os_i - \frac{1}{N^+} \sum_j \mathbb{1}(os_j > \tau)os_j]^2 + \frac{1}{N^-} \sum_i [os_i - \frac{1}{N^-} \sum \mathbb{1}(os_j \leq \tau)os_j]^2
\tag{10}
$$

To ensure a stable estimation of the outlier distribution, the distribution is updated using an exponential moving average manner with a length of $N_a$. Here, it ranges from 0 to 1, and the step size is set to 0.01. Upon confirming the effective feature extraction of class samples, resulting in $\mathcal{F}'(x_i)$ and $\mathcal{F}'(x'_i)$, we obtain the feature distribution $\mathcal{D}_s$ of the weak OOD in the source domain, ascertained during the pre-TTT stage. For handling weak OOD samples, we employ a strategy that integrates the contrastive learning loss NT-XENT with negative log-likelihood loss. This approach aims to embed the test sample $x_i$ nearer to the cluster center of its respective class while distancing it from the cluster centers of other classes. The formulation of the negative log-likelihood loss is detailed below:

$$
\mathcal{L}_{PC}^{wea} = - \sum_{k \in Y_s} \mathbb{1}(\hat{y} = k) \log \frac{\exp(\frac{<d_k, \mathcal{F}'(x_i)>}{\delta})}{\sum_l \exp(\frac{<d_l, \mathcal{F}'(x_i)>}{\delta})}
\tag{11}
$$

where $\delta$ is a hyper-parameter, set to 0.1 in all experiments.

To enhance the robustness of sample classification and streamline computation, the feature distribution for the current batch has been quantified based on pseudo-labels $\hat{y} = k$. The corresponding formula is articulated as follows:

$$
d_k^c = \frac{1}{2K} \sum_{i=1}^{K} (\mathcal{F}'(x) + \mathcal{F}'(x'))
\tag{12}
$$

In the current batch, there are $k$ sample pairs in class $K$, and their average feature distribution is $d_k^c$. Initially, positive sample pairs are normalized employing the L2 norm. The specific formula utilized for this normalization is detailed below:

$$v_i^c = \frac{d_i^c}{\sqrt{\sum_{i=1}^M (d_i^c)^2}},$$
$$v_i^s = \frac{d_i^s}{\sqrt{\sum_{i=1}^M (d_i^s)^2}} \tag{13}$$

Using normalized vectors $v_i^c$ and $v_i^s$, the NT-XENT loss is computed:

$$\mathcal{L}_{NT} = -\alpha_2 (\log(\frac{\mathcal{S}(v_i^c, v_j^s)_{pos}}{\sum_{k \neq i}^M \mathcal{S}(v_k^c, v_j^s)_{neg} + \mathcal{S}(v_i^c, v_j^s)_{pos}})$$
$$+ \log(\frac{\mathcal{S}(v_i^c, v_j^s)_{pos}}{\sum_{k \neq j}^M \mathcal{S}(v_i^c, v_k^s)_{neg} + \mathcal{S}(v_i^c, v_j^s)_{pos}})) \tag{14}$$

$\alpha_2$ adjusts the loss's impact magnitude. The similarity computation incorporates a temperature normalization factor $\gamma_2$, pivotal in adjusting the scale of similarity measures within the model.

For categorizing samples as strong OOD, the following conditions or mathematical criteria must be met:

$$\hat{os}_i = 1 - \max_{d_k \in \mathcal{D}_s \cup \mathcal{D}_{str}} \langle \mathcal{F}'(x_i), d_k \rangle \tag{15}$$

When strong OOD samples fulfill a certain criterion, they are incorporated into the existing strong OOD class. If not, a new strong OOD cluster center is established. In the real-world application of machine learning models, the classes known and trained on in the source domain are finite and predetermined. However, the emergence of new classes in practical scenarios is theoretically infinite. To prevent the unbounded growth of OOD cluster centers, the distribution $\mathcal{D}_{str}$ is managed as a queue with a fixed capacity of $N_q$. The value of $N_q$ is 100. As new OOD prototypes are introduced, the oldest prototypes are phased out.

Concurrently, the negative log-likelihood loss for these samples is computed as follows:

$$\mathcal{L}_{PC}^{str} = -\sum_{k \in Y_{str}} \mathbb{1}(\hat{y} = k) \log \frac{\exp(\frac{<d_k, \mathcal{F}'(x_i)>}{\delta})}{\sum_l \exp(\frac{<d_l, \mathcal{F}'(x_i)>}{\delta})} \tag{16}$$

Self-training (ST) is susceptible to the issue of incorrect pseudo-labels, known as confirmation bias. This self-supervised confirmation bias can exacerbate over time, significantly impacting performance. Particularly in the presence of strong OOD samples within the target domain, the model may erroneously classify these as belonging to known categories, even with low confidence, thereby intensifying the confirmation bias. To mitigate the risk of ST failure, we adopt distribution alignment as a form of self-training regularization, drawing on insights from previous studies. This approach aims to reduce the adverse effects of confirmation bias by ensuring that the model's predictions are more aligned with the actual distribution of the data.

The features in the source domain are assumed to follow a Gaussian distribution $\mathcal{N}(\mu_s, \sum_s)$. In the target domain, the feature distribution $\mathcal{N}(\mu_t, \sum_t)$ is estimated using a momentum parameter $\beta$, incorporating only test samples pruned via strong OOD criteria. To refine clustering in the target domain, we use the Kullback-Leibler Divergence loss $L_{KLD}$:

$$\mathcal{L}_{KLD} = D_{KL}(\mathcal{N}(\mu_s, \sum_s) || \mathcal{N}(\mu_t, \sum_t)) \tag{17}$$

For the sake of aesthetics, we have simplified the formula. As a result, the final loss function for the phase of contrastive alignment by cluster centers and sample pairs can be articulated as follows:

$$\mathcal{L}_{cs} = \mathcal{L}_{NT} + \mathcal{L}_{PC}^{wea} + \mathcal{L}_{PC}^{str} + \mathcal{L}_{KLD}$$

$$= -\alpha_2 (\log(\frac{\mathcal{S}(v_i^c, v_j^s)_{pos}}{\sum_{k \neq i}^M \mathcal{S}(v_k^c, v_j^s)_{neg} + \mathcal{S}(v_i^c, v_j^s)_{pos}}) + \log(\frac{\mathcal{S}(v_i^c, v_j^s)_{pos}}{\sum_{k \neq j}^M \mathcal{S}(v_i^c, v_k^s)_{neg} + \mathcal{S}(v_i^c, v_j^s)_{pos}}))$$

$$- (\sum_{k \in Y_s} \mathbb{1}(\hat{y} = k) \log \frac{\exp(\frac{<d_k, \mathcal{F}'(x_i)>}{\delta})}{\sum_l \exp(\frac{<d_l, \mathcal{F}'(x_i)>}{\delta})} + \sum_{k \in Y_{str}} \mathbb{1}(\hat{y} = k) \log \frac{\exp(\frac{<d_k, \mathcal{F}'(x_i)>}{\delta})}{\sum_l \exp(\frac{<d_l, \mathcal{F}'(x_i)>}{\delta})})$$

$$+ D_{KL}(\mathcal{N}(\mu_s, \textstyle\sum_s) || \mathcal{N}(\mu_t, \textstyle\sum_t))$$

$$\tag{18}$$

---

**Algorithm 1: OWDCL algorithm.**

---

**Input:**
1) Source domain data $\mathcal{X}_s$
2) Target domain data $\mathcal{X}_t$
3) Pre-trained model $\mathcal{F}(\cdot)$

**1** Utilize $\mathcal{X}_s$ with $\mathcal{F}(\cdot)$ to obtain source label features $\mathcal{D}_s = \{d_1^s, ..., d_m^s\}$ as shown in Eq. 1
**2** Initialize all parameters
**4 for** $l \leftarrow 0$ to $L$ **do**
**6**     Randomly sample a batch of data $x$ from $\mathcal{X}_t$.
**8**     Apply data augmentation to $x$ to obtain augmented counterparts $x'$.
**10**     Use $\mathcal{F}(\cdot)$ to extract features $\mathcal{F}(x)$ and $\mathcal{F}(x')$.
**12**     Utilize Eq. 7 to compute $\mathcal{L}_{ps}$, enhancing the performance of $\mathcal{F}(\cdot)$.
**14**     Calculate cosine similarity between $\mathcal{F}(x)$ and $\mathcal{F}(x')$ for each known class $d_m^s$ using Eq. 8
      and clustering.
**16**     Generate class predictions based on clustering results and compute
      $\mathcal{L}_{NT}, \mathcal{L}_{PC}^{wea}, \mathcal{L}_{PC}^{str}, \mathcal{L}_{KLD}$.
**18**     Train $\mathcal{F}(\cdot)$ using the losses $\mathcal{L}_{NT}, \mathcal{L}_{PC}^{wea}, \mathcal{L}_{PC}^{str}, \mathcal{L}_{KLD}$, and $\mathcal{L}_{ps}$.
**19 end**
**20 return** $\mathcal{F}(\cdot)$

---

# 3 EXPERIMENTS

## 3.1 DATASETS AND EVALUATION METRIC

Several datasets are utilized to fully demonstrate the validity of our method. For the corruption datasets, we use the following datasets, CIFAR10-C/CIFAR100-C Hendrycks & Dietterich (2019), each containing 10000 corrupt images with 10/100 classes, and ImageNet-C Hendrycks & Dietterich (2019), which contains 5000 corrupt images within 1000 classes. For the style transfer dataset, we introduce the Tiny-ImageNet Le & Yang (2015) consists of 200 classes with each class containing 500 training images and 50 validation images. For other common datasets, We also introduce MNIST LeCun et al. (1998) is a handwritten digit dataset, that contains 60,000 training images and 10,000 testing images. SVHN Netzer et al. (2011) is a digital dataset in a real street context, including 50,000 training images and 10,000 testing images.

To evaluate open-world test-time training, we adopt the same evaluation metric as OWTTT Li et al. (2023). To set up a fair comparison with existing methods, we take all the classes in the TTT benchmark dataset as seen classes and add additional classes from additional datasets as unseen classes. In the later experiments, we set the number of known class samples and the number of unknown class samples to be the same. Then we follow the "One Pass" protocol Su et al. (2022), Firstly, the training objective cannot be changed during the source domain training procedure. Secondly, testing data in the target domain is sequentially streamed and predicted. In this problem, we evaluate whether we can judge the accuracy of the source domain class as a strong OOD. First, the accuracy of the source domain class is recorded as $Acc_S$:

Table 1: Open-world test-time training results on CIFAR10-C. All values are presented in percentages (%), with the best results highlighted in bold.

| Method | Noise | | | MNIST | | | SVHN | | |
|---|---|---|---|---|---|---|---|---|---|
| | $Acc_S$ | $Acc_N$ | $Acc_H$ | $Acc_S$ | $Acc_N$ | $Acc_H$ | $Acc_S$ | $Acc_N$ | $Acc_H$ |
| TEST | 68.59 | 99.97 | 81.36 | 60.48 | 88.81 | 71.96 | 60.94 | 86.44 | 71.48 |
| BNIoffe & Szegedy (2015) | 76.63 | 95.69 | 85.11 | 76.15 | 95.75 | 84.83 | 79.18 | 94.71 | 86.25 |
| TTT++Liu et al. (2021c) | 41.09 | 57.31 | 47.86 | 59.52 | 77.52 | 67.34 | 68.77 | 85.80 | 76.34 |
| TENTWang et al. (2020) | 32.24 | 33.30 | 32.77 | 55.64 | 68.27 | 61.31 | 66.70 | 82.50 | 73.77 |
| SHOTLiang et al. (2020) | 63.54 | 71.37 | 67.23 | 56.92 | 53.26 | 55.03 | 70.01 | 72.58 | 71.27 |
| TTACSu et al. (2022) | 64.46 | 77.42 | 70.35 | 77.60 | 84.53 | 80.92 | 77.30 | 81.10 | 79.16 |
| OWTTTLi et al. (2023) | 85.46 | 98.60 | 91.56 | 83.89 | 97.83 | 90.32 | 84.99 | 87.94 | 86.44 |
| OWDCL(Ours) | **87.16** | **99.99** | **93.08** | **85.59** | **99.14** | **91.82** | **85.35** | **89.74** | **87.49** |

Table 2: Open-world test time training results on CIFAR100-C. All values are presented in percentages (%), with the best results highlighted in bold.

| Method | Noise | | | MNIST | | | SVHN | | |
|---|---|---|---|---|---|---|---|---|---|
| | $Acc_S$ | $Acc_N$ | $Acc_H$ | $Acc_S$ | $Acc_N$ | $Acc_H$ | $Acc_S$ | $Acc_N$ | $Acc_H$ |
| TEST | 36.75 | 99.87 | 53.73 | 25.99 | 49.59 | 34.11 | 30.01 | 81.62 | 43.89 |
| BNIoffe & Szegedy (2015) | 50.21 | 98.72 | 66.56 | 36.21 | 84.69 | 50.73 | 45.69 | 90.45 | 60.71 |
| TTT++Liu et al. (2021c) | 23.47 | 70.26 | 35.19 | 28.31 | **86.74** | 42.68 | 37.56 | 90.45 | 53.08 |
| TENTWang et al. (2020) | 22.57 | 66.60 | 33.72 | 27.85 | 80.92 | 41.43 | 37.08 | 89.90 | 52.51 |
| SHOTLiang et al. (2020) | 51.52 | 98.21 | 67.58 | 35.35 | 81.71 | 49.35 | 45.87 | 89.72 | 60.70 |
| TTACSu et al. (2022) | 51.11 | 98.66 | 67.34 | 37.78 | 86.66 | 52.62 | 47.29 | **91.42** | 62.33 |
| OWTTTLi et al. (2023) | 56.76 | 97.25 | 71.68 | 40.77 | 82.91 | 54.66 | 54.32 | 81.98 | 65.34 |
| OWDCL(Ours) | **58.20** | **99.93** | **73.23** | **44.01** | 81.85 | **56.69** | **55.38** | 82.80 | **66.36** |

Table 3: Open-world test time training results on ImageNet-C. All values are presented in percentages (%), with the best results highlighted in bold.

| Method | Noise | | | MNIST | | | SVHN | | |
|---|---|---|---|---|---|---|---|---|---|
| | $Acc_S$ | $Acc_N$ | $Acc_H$ | $Acc_S$ | $Acc_N$ | $Acc_H$ | $Acc_S$ | $Acc_N$ | $Acc_H$ |
| TEST | 18.51 | **100.00** | 31.24 | 18.66 | 98.27 | 31.36 | 18.94 | 87.75 | 31.15 |
| BNIoffe & Szegedy (2015) | 36.34 | 99.97 | 53.31 | 30.77 | 74.53 | 43.55 | 33.26 | 84.54 | 47.74 |
| TENTWang et al. (2020) | 22.54 | 10.47 | 14.29 | 27.53 | 10.01 | 14.68 | 41.16 | 45.51 | 43.22 |
| SHOTLiang et al. (2020) | **46.79** | **100.00** | **63.75** | 27.47 | 55.25 | 36.70 | 34.00 | 75.94 | 46.97 |
| TTACSu et al. (2022) | 42.60 | 94.52 | 58.73 | 30.43 | 72.11 | 42.80 | 31.59 | 74.07 | 44.29 |
| OWTTTLi et al. (2023) | 41.40 | **100.00** | 58.56 | 38.86 | 93.35 | 54.87 | 38.60 | 98.06 | 55.40 |
| OWDCL(Ours) | 41.96 | **100.00** | 59.11 | **41.70** | **99.92** | **57.00** | **42.23** | **99.25** | **57.70** |

$$Acc_S = \frac{\sum_{x_i,y_i \in \mathcal{D}_t} \mathbb{1}(y_i = \hat{y}_i) \cdot \mathbb{1}(y_i \in \mathcal{C}_s)}{\sum_{x_i,y_i \in \mathcal{D}_t} \mathbb{1}(y_i \in \mathcal{C}_s)} \quad (19)$$

This is followed by the rejection of strong OOD, which successfully rejects the accuracy of the strong OOD sample and is recorded as $Acc_N$:

$$Acc_N = \frac{\sum_{x_i,y_i \in \mathcal{D}_t} \mathbb{1}(y_i \in \mathcal{C}_t \setminus \mathcal{C}_s) \cdot \mathbb{1}(y_i \in \mathcal{C}_t \setminus \mathcal{C}_s)}{\sum_{x_i,y_i \in \mathcal{D}_t} \mathbb{1}(y_i \in \mathcal{C}_t \setminus \mathcal{C}_s)} \quad (20)$$

And finally, their tradeoff, set to $Acc_H$:

$$Acc_H = 2 \cdot \frac{Acc_S \cdot Acc_N}{Acc_S + Acc_N} \quad (21)$$

where $\hat{y}_i$ refers to the predicted label and $\mathbb{1}(y_i \in \mathcal{C}_s)$ is true if $y_i$ is in the set $\mathcal{C}_s$.

### 3.2 EXPERIMENTAL ANALYSIS

#### 3.2.1 ABLATION STUDY

In our extensive ablation study conducted on the CIFAR10-C dataset, we incorporated Noise as a representative of strong OOD scenarios, alongside 15 different types of corruption present in the

Table 4: Model ablation experiment

| $\mathcal{PS}$ | $\mathcal{CS}$ | $Acc_S$ | $Acc_N$ | $Acc_H$ |
|---|---|---|---|---|
| ✗ | ✗ | 85.46 | 98.60 | 91.56 |
| ✔ | ✗ | 86.54 | **99.99** | 92.78 |
| ✗ | ✔ | 86.89 | **99.99** | 92.93 |
| ✔ | ✔ | **87.16** | **99.99** | **93.08** |

original dataset. Due to constraints in length, we present the final averaged results; the details of which are illustrated in Table 4. In this study, $\mathcal{PS}$ denotes the enhancements made in the Contrastive Alignment by Positive Sample Pairs segment, and $\mathcal{CS}$ signifies the advancements in the Contrastive Alignment by Cluster and Sample Pairs aspect. The baseline, denoted as OWTTT, does not incorporate any of these improvements. Our findings indicate that each improvement significantly outperforms the baseline. This achievement is particularly notable in effectively differentiating strong OOD while simultaneously accurately classifying weak OOD.

### 3.2.2 COMPARISON SETTINGS

For all competing methods that are set by default, we equip them with the same strong OOD detector introduced in Li et al. (2023). For all models, ResNet-50 He et al. (2016) was selected as the backbone, SGD was selected as the optimizer, and the learning rate was set to 0.01/0.001 and batch size to 256 in CIFAR10-C/CIFAR100-C. In ImageNet-C, the learning rate is set to 0.001 and the batch size is set to 128. The other hyperparameter Setting of the model refer to the default Settings of the original paper. For the data enhancement of the positive sample of OWDCL(ours), we only perform rotation in order (0-30 degrees), flipping horizontally. Because of the noise effect of domain shift, combined with overly complex data enhancement, it will make the model difficult to fit.

For the CIFAR10-C/CIFAR100-C datasets, the hyperparameters are configured as follows: $\gamma_1$ is set to 0.8, $\gamma_2$ to 0.4, $\alpha_1$ to 1, and $\alpha_2$ to 2. In the ImageNet-C dataset, both $\gamma_1$ and $\gamma_2$ are uniformly set at 1. Regarding $\alpha_1$, initially set at 1, we reduce it to 0.1 after the 20th batch to mitigate potential overfitting issues identified in more complex datasets, where $\mathcal{L}_{ps}$ remains impactful in the initial stages. Regarding the other parameters, their settings are consistent throughout the document and were initially introduced at their first mention. These specific configurations draw upon established practices from previous research Li et al. (2023).

### 3.2.3 COMPARATIVE EXPERIMENTS

We first evaluate open-world test-time training under noise corrupted target domain. We treat CIFAR10/CIFAR100 Krizhevsky et al. (2009) and ImageNet Deng et al. (2009) as the source domain and test-time adapt to CIFAR10-C, CIFAR100-C, and ImageNet-C as the target domain respectively.

For experiments on CIFAR10/100, we introduce random noise, MNIST, SVHN, Tiny-ImageNet with non-overlap classes, and CIFAR100 as strong OOD testing samples. Table 6 compares the classification error of our proposed method against recent TTT methods on the CIFAR10-C dataset. Table 7 shows the performance comparison results on the CIFAR100-C dataset. It can be seen that for different strong OOD, our models have shown extremely excellent performance, and basically, under each strong OOD, our accuracy has been improved by more than 2%. In the CIFAR10-C dataset, we added Tiny-ImageNet as a strong OOD, which improved our accuracy by nearly 5% for this complex strong OOD.

In CIFAR100-C, due to the complexity of data set categories and the interference of strong OOD, many models have significantly improved the recognition accuracy of strong OOD ($ACC_N$). However, his weak OOD ($ACC_S$) accuracy drops sharply, which is caused by stong OOD interference, and he loses the ability to recognize the source domain classes. OWDCL not only demonstrates significant performance improvements compared to traditional TTT models but also incorporates contrastive learning to enhance the model's feature extraction capabilities. This enhancement helps to prevent the misclassification of weak OOD samples as strong OOD by improving feature extraction. Compared to OWTTT, OWDCL generally achieves an accuracy improvement of about 1-4%,

highlighting the effectiveness of integrating contrastive learning for more robust feature discrimination and OOD handling.

For ImageNet-C, we introduce random noise, MNIST, and SVHN as strong OOD samples. Very encouraging results are also obtained on the large-size complicated ImageNet-C dataset, as shown in Table 3. Our model shows a similar effect for large data sets. For random noise as strong OOD, our method is inferior to SHOT. We believe that random noise prevents us from extracting features from strong OOD, thus affecting the final performance. In experiments where MNIST and SVHN were used as strong OOD samples, our OWDCL model's classification accuracy for weak OOD ($ACC_S$) increased by approximately 4% compared to OWTTT, a more pronounced improvement than observed with the CIFAR10-C/CIFAR100-C datasets. This suggests that the complexity of the dataset significantly impacts the model's feature extraction requirements, making weak OOD samples more susceptible to being misclassified as strong OOD. Our method's enhancements effectively address this issue, demonstrating that the more complex the dataset, the more pronounced the benefits of our model become.

Finally, our proposed method consistently outperforms all competing methods under most experiment settings, suggesting the effectiveness of the proposed method.

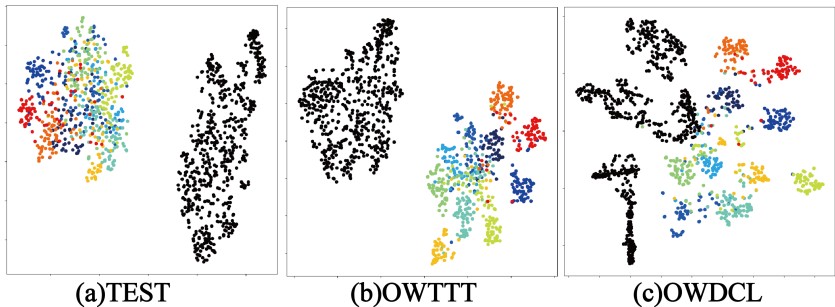

| (a)TEST | (b)OWTTT | (c)OWDCL |

Figure 3: Visual analysis experiment. Black is strong OOD, while the others are weak OOD.

### 3.2.4 VISUALIZED ANALYSIS

We conducted a visual analysis on the CIFAR10-C dataset, using Gaussian noise as the corruption factor and the MNIST dataset as the benchmark for strong OOD scenarios. Three models - TEST, OWTTT, and OWDCL - were assessed using data from their last five batches. This data underwent dimensionality reduction via t-SNE, followed by a subsequent visualization. In these visualizations, black indicates the strong OOD class, while ten other colors represent the ten CIFAR-10 classes, as detailed in Figure 3. Compared to TEST, OWTTT showed improved classification accuracy but with a significantly higher misclassification rate. OWDCL further excelled by enlarging the spatial separation between distinct classes, indicating superior performance. Notably, OWDCL demonstrated remarkable feature extraction capabilities for unknown strong OODs during the Test-Time Training (TTT) process, despite being initially trained on MNIST. This ability is evidenced by the emergence of distinct class clusters, even though it does not precisely classify each of the ten MNIST classes.

## 4 CONCLUSION

In this paper, we introduce a novel method called Open World Dynamic Contrastive Learning (OWDCL), which effectively addresses the limitations of traditional Test-Time Training (TTT) methods in open-world scenarios. By creatively leveraging contrastive learning to generate positive sample pairs, OWDCL significantly enhances initial feature extraction and reduces the misclassification of weak OOD data as strong OOD. This methodology not only improves discriminability in the early stages of TTT but also strengthens the overall robustness of the model against strong OOD data. With superior performance across various datasets, OWDCL establishes a new benchmark in the field of Open-World Test-Time Training.

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

## A  APPENDIX

### A.1  RELATED WORK

#### A.1.1  UNSUPERVISED DOMAIN ADAPTATION

Unsupervised domain adaptation (UDA) Ganin & Lempitsky (2015); Wang & Deng (2018); Liu et al. (2022) aims to adapt models trained on a source domain to unlabeled target domain data. UDA typically employs strategies like difference loss Long et al. (2015), adversarial training Ganin & Lempitsky (2015), and self-supervised training Liu et al. (2021a) to learn invariant properties across domains. Despite considerable progress in enhancing target domain generalizability, UDA's reliance on both source and target domains during adaptation is often impractical, e.g., due to data privacy concerns. Consequently, source-free domain adaptation Xia et al. (2021); Liu et al. (2021b); Yang et al. (2021); Kundu et al. (2020) has emerged, eliminating the need for source domain data and relying solely on a pre-trained model and target domain data.

Table 5: Characteristics of problem settings that adapt a trained model to a potentially shifted test domain. 'Offline' adaptation assumes access to the entire source or target dataset, while 'Online' adaptation can automatically predict a single or batch of incoming test samples.

| Setting | Source | Target | Train Loss | Test Loss | Offline | Online | Strong OOD |
|---|---|---|---|---|---|---|---|
| Fine-tuning | ✗ | $x^t, y^t$ | $\mathcal{L}(x^s, y^s)$ | - | ✔ | ✗ | ✗ |
| Unsupervised Domain Adaptation | $x^s, y^s$ | $x^t$ | $\mathcal{L}(x^s, y^s) + \mathcal{L}(x^s, x^t)$ | - | ✔ | ✗ | ✗ |
| Universal Domain Adaptation | $x^s, y^s$ | $x^t$ | $\mathcal{L}(x^s, y^s) + \mathcal{L}(x^s)$ | - | ✔ | ✗ | ✔ |
| Domain Generalization | $x^s, y^s$ | ✗ | $\mathcal{L}(x^s, y^s)$ | - | ✔ | ✗ | ✗ |
| Source-free Domain Adaptation | ✗ | $x^t$ | $\mathcal{L}(x^s, x^t)$ | - | ✔ | ✗ | ✗ |
| Test-time training(TTT) | $x^s, y^s$ | $x^t$ | $\mathcal{L}(x^s, y^s) + \mathcal{L}(x^s)$ | $\mathcal{L}(x^t)$ | ✗ | ✔ | ✗ |
| Test-time adaptation(TTA) | ✗ | $x^t$ | ✗ | $\mathcal{L}(x^t)$ | ✗ | ✔ | ✗ |
| Open-World Test-time training(OWTTT) | $x^s, y^s$ | $x^t$ | $\mathcal{L}(x^s, y^s) + \mathcal{L}(x^s)$ | $\mathcal{L}(x^t)$ | ✗ | ✔ | ✔ |

#### A.1.2  TEST-TIME TRAINING

In scenarios requiring adaptation to arbitrary unknown target domains with low inference latency and without source domain data access, Test-Time Training/Adaptation (TTT/TTA) Liu et al. (2021c); Wang et al. (2020); Sun et al. (2020) has emerged as a new paradigm. TTT/TTA can be achieved not only by adjusting model weights to align features with the source domain distribution Liu et al. (2021c); Su et al. (2022) but also through self-training that reinforces model predictions on unlabeled data Wang et al. (2020); Chen et al. (2022); Niu et al. (2022). However, TTT/TTA, limited by the absence of target domain labels, often relies on summarizing the target domain's feature distribution to approximate and align with the correct source domain distribution, enhancing model performance. This approach, while reducing uncertainty, is prone to errors, especially under strong OOD interference in open-world scenarios Li et al. (2023).

#### A.1.3  OPEN-SET DOMAIN ADAPTATION

To address open-world scenarios, Open-Set Domain Adaptation (OSDA) has been proposed Panareda Busto & Gall (2017). Existing OSDA methods include strategies like transforming logits of unknown class samples into a recognizable constant Saito et al. (2018), and defining and maximizing the distance between open-set and closed-set Panareda Busto & Gall (2017). Additionally, Universal Adaptation Network (UAN) approaches consider scenarios where unknown classes exist in both source and target domains You et al. (2019). Further, in scenarios lacking access to source domain data, Universal source-free Domain Adaptation has been explored Kundu et al. (2020). There is very poor research on open-world test-time training (OWTTT) Li et al. (2023). There is a lack of research to solve the problem of weak OOD accuracy due to the lack of feature extraction ability in the initial model.

### A.2  COMPARISON METHODS AND SETTINGS

Given that open-world Test-Time Training (OWTTT) is a relatively unexplored area with limited studies, our comparison necessarily includes other Test-Time Training (TTT) models, drawing on insights from previous research. It's important to note that while TTT is a method optimized for

real-time testing, it differs from test-time adaptation in that it utilizes parts of the source domain data, such as small batch samples or source domain BN layer statistics, under real-time constraints. This includes the feature distribution of the source domain, as seen in OWTTT and our OWDCL model. Therefore, including traditional TTT models in our experimental comparison is justified. Our comparison model is as follows:

**TEST**: Evaluating the source domain model on testing data.

**BN** Ioffe & Szegedy (2015): Updating batch norm statistics on the testing data for test-time adaptation.

**TTT++** Liu et al. (2021c): Aligns source and target domain distribution by minimizing the F-norm between the mean covariance.

**TENT** Wang et al. (2020): This method fine-tunes scale and bias parameters of the batch normalization layers using an entropy minimization loss during inference.

**SHOT** Liang et al. (2020): Implements test-time training by entropy minimization and self-training. SHOT assumes the target domain is class balanced and introduces an entropy loss to encourage uniform distribution of the prediction results.

**TTAC** Su et al. (2022): Employs distribution alignment at both global and class levels to facilitate test-time training.

**OWTTT** Li et al. (2023): Which combines self-training with prototype expansion to accommodate the strong OOD samples.

Table 6: Open-world test time training results on CIFAR10-C. All values are presented in percentages (%), with the best results highlighted in bold.

| Method | Tiny-ImageNet | | | CIFAR100-C | | |
|---|---|---|---|---|---|---|
| | $Acc_S$ | $Acc_N$ | $Acc_H$ | $Acc_S$ | $Acc_N$ | $Acc_H$ |
| TEST | 57.41 | 79.63 | 66.72 | 52.74 | 74.24 | 61.67 |
| BNIoffe & Szegedy (2015) | 67.66 | 82.67 | 74.42 | 68.44 | 81.38 | 74.35 |
| TTT++Liu et al. (2021c) | 66.70 | 79.28 | 72.44 | 65.69 | 77.47 | 71.10 |
| TENTWang et al. (2020) | 66.54 | 79.32 | 72.37 | 64.80 | 76.40 | 70.12 |
| SHOTLiang et al. (2020) | 67.78 | 82.25 | 74.32 | 67.73 | 72.87 | 70.21 |
| TTACSu et al. (2022) | 71.64 | 77.14 | 74.29 | 71.94 | 75.44 | 73.65 |
| OWTTTLi et al. (2023) | 71.77 | 84.71 | 77.70 | 74.08 | 84.64 | 79.01 |
| OWDCL(Ours) | **76.57** | **86.34** | **81.20** | **78.47** | **85.47** | **81.82** |

Table 7: Open-world test time training results on CIFAR100-C. All values are presented in percentages (%), with the best results highlighted in bold.

| Method | Tiny-ImageNet | | | CIFAR10-C | | |
|---|---|---|---|---|---|---|
| | $Acc_S$ | $Acc_N$ | $Acc_H$ | $Acc_S$ | $Acc_N$ | $Acc_H$ |
| TEST | 25.41 | 70.06 | 37.30 | 25.55 | 73.28 | 37.89 |
| BNIoffe & Szegedy (2015) | 34.88 | **82.18** | 48.97 | 37.00 | 83.54 | 51.28 |
| TTT++Liu et al. (2021c) | 34.67 | 81.25 | 48.60 | 33.78 | 81.12 | 47.70 |
| TENTWang et al. (2020) | 35.51 | 77.34 | 48.60 | 35.20 | 80.26 | 48.94 |
| SHOTLiang et al. (2020) | 35.72 | 81.11 | 49.59 | 38.00 | 82.13 | 51.96 |
| TTACSu et al. (2022) | 32.04 | 80.46 | 45.83 | 38.83 | 83.68 | 53.05 |
| OWTTTLi et al. (2023) | 38.90 | 81.92 | 52.75 | 38.97 | 83.20 | 53.08 |
| OWDCL(Ours) | **40.91** | 81.53 | **54.48** | **41.46** | **83.73** | **55.46** |

### A.3 FURTHER PERFORMANCE ANALYSIS

### A.3.1 LOSS CONVERGENCE AND ACCURACY ANALYSIS

In our experiments on the CIFAR10-C dataset, we use Noise as the Strong OOD corruption and record the loss convergence and $ACC_H$ accuracy trends for each batch, As shown in Figure 4, we present the model's performance under four randomly selected Weak OOD corruption types. The results highlight the model's capacity to adapt and converge under various corruption scenarios, demonstrating its robustness in handling OOD samples and its accuracy on the most challenging corrupted data.

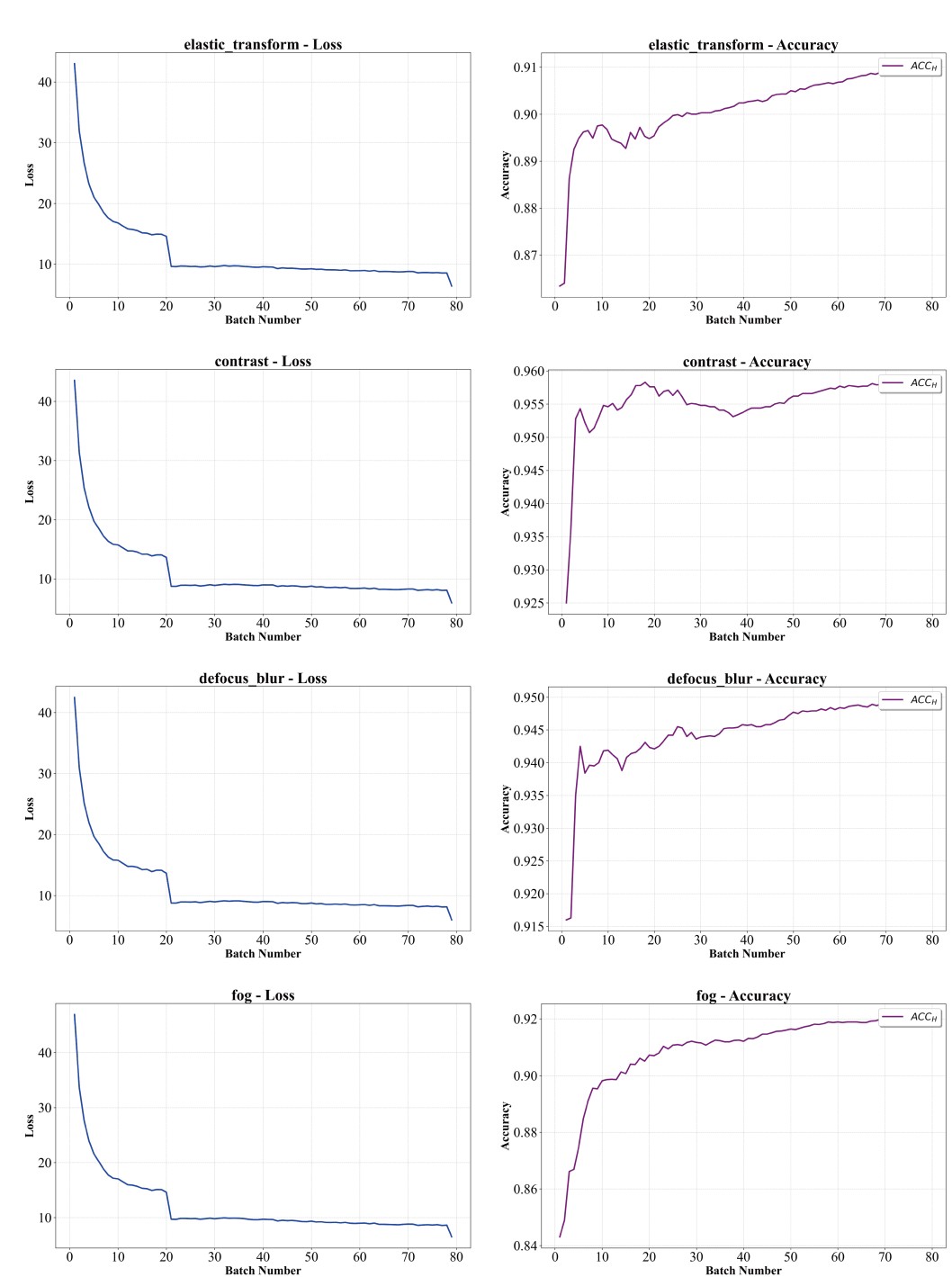

Figure 4: Open-world Test Time Training on CIFAR10-C: Loss Convergence and Accuracy $ACC_H$ under Noise (Strong OOD)

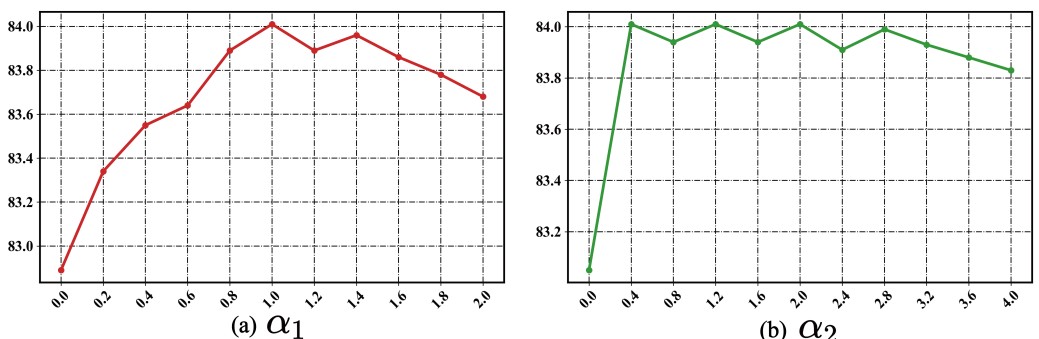

Figure 5: Parameter Robustness Analysis.

### A.3.2 PARAMETER ROBUSTNESS ANALYSIS

In the context of parameter settings for the experiment, our approach OWDCL, being an extension of OWTTT, refers to the parameter configuration of OWTTT, adhering to a consistent parameter setup throughout the paper. Owing to the numerous secondary parameters involved in our method, the specific design values were mentioned in their initial introduction, and a unified approach was adopted for all experiments. In the parameter robustness analysis, we scrutinized the primary parameters $\alpha_1$ and $\alpha_2$ to evaluate their robustness. The experiments were conducted under the Noise condition in the CIFAR10-C dataset, as depicted in Figure 5. From the illustration, it is evident that the model's accuracy maintains commendable performance within a certain range, thus affirming the robustness of our two parameters over a defined interval.

