# OpenReview forum: "Open-World Test-Time Training: Self-Training with Contrastive Learning"
_ICLR.cc/2025/Conference — ICLR 2025 Conference Withdrawn Submission_

### Official Review · Reviewer_15Hh · 2024-10-17

**Soundness:** 2
**Presentation:** 1
**Contribution:** 2
**Rating:** 3
**Confidence:** 3

**Summary:**

The paper presents a new approach for open-world test-time training (OWTTT). The main challenge faced with OWTTT approaches is their sensitivity to (even mild) distribution shift. To resolve this issue, previous work focused on improving the ability to distinguish samples with weak distribution shift and samples with strong shift. This work, on the other hand, proposes instead a self-supervised approach where the goal is to improve performance on weak OOD samples. Specifically, the others propose a contrastive loss for training the model, which improves performances on samples with, for example, different contrast.

**Strengths:**

- The paper proposes an interesting change to the TTT pipeline in order to improve generalization.
- The results of the paper are encouraging on some vision distribution shifts.
- The proposed approach is simple and seems effective.

**Weaknesses:**

- The writing could be improved. *1)* In-text citations are missing parentheses. *2)* The authors spend a lot of time explaining $\ell_2$-norm and other basic definitions, but don't provide a preliminary on test-time training, which is more interesting and requires some explanation. *3)* There a bunch of typos all over the paper and missed dots.
- The results, although seem to improve, are still very close. Would be interesting to measure the standard deviation of model performance as opposed to just the mean.
- The evaluation leverages basic OOD datasets, such as CIFAR-10C and ImageNet-C. There exists other benchmarks that could be used and that are more interesting (other ImageNet variations, WILDS, etc.)

**Questions:**

Check weaknesses.

---

### Official Review · Reviewer_gc2G · 2024-10-23

**Soundness:** 3
**Presentation:** 2
**Contribution:** 3
**Rating:** 5
**Confidence:** 4

**Summary:**

This paper incorporates the idea of contrastive learning into OWTTT [1], propose a novel method abbreviated as OWDCL, which attempts to tackle the problem introduced by [1]. More specifically, The pipeline of [1] focuses on distinguishing between strong and weak OOD through expanding prototypes dynamically to handle Open-World Out-Of-Distribution (OWOOD) scenario, while may misclassifying the weak OOD data as strong OOD noise, causing feature extraction ability for weak OOD is hampered. The experiment results of  this paper demonstrated OWDCL can recover the reprehensive ability under OWOOD scenario to some extent.

References
- [1] Li, Y., Xu, X., Su, Y., & Jia, K. (2023). On the Robustness of Open-World Test-Time Training: Self-Training with Dynamic Prototype Expansion. 2023 IEEE/CVF International Conference on Computer Vision (ICCV), 11802-11812.

**Strengths:**

**Strengths**
- The idea of adding an attention mechanism to recognize hard samples is concise and intuitive.
- The experiments are comprehensive and strong enough to support the efficiency of the proposed idea, they demonstrate that introducing the idea of contrastive learning  can definitely introduce significant improvements across various OWOOD scenarios.

**Weaknesses:**

**Weaknesses**
- There are a lot of typos in this paper, for example,

    - Substitute $\mathcal{F}$ to $\mathcal{F}^\prime$ in (4).
    - Rewrite the inner product of positive pair as $v_i \cdot v_j^\prime$ or $v_i^Tv^\prime_j$ in (6).
    - $i$ should be subscript in (9).
    - $\hat{y}_i$ is more easily understood expression in (11).
    - The summation component does not rely on $i$ in the RHS of (12)?
- The paragraph under (15) is not clear enough. Referring to [1] may be a great way to improve readability.  The overall readability of the paper is not very good, authors should carefully consider whether the use of notation is appropriate to improve the readability of the article, especially for usage of subscripts, like replacing $d_m$ into $d^m_i$, $M$ into $n_s^i$ in (1), giving the specific formula for $K$ of (12) are a little bit better, directly comparing the pipelines of OWDCL and OWTTT in a certain paragraph can help readers with relevant field foundations quickly understand the contribution of this paper.
- I can't understand why the authors of both OWTTT and this paper adopt the criteria of $Acc_N$. This quantity measures the goodness of recognizing strong OOD rather than the accuracy of strong OOD. In practice, it is possible to calculate the classification error for strong OOD data, isn't it?

**Questions:**

**Questions**
- Comparing to [1], the main additional idea of this paper is adding the term of contrastive loss to enhance the final model performance, especially within weak OOD classes. If your intuition is correct, does it imply that the pretrained representation through self-supervised learning method based on data augmentation, combined with the same pipeline according to [1], instead of adopting supervised pretraining techniques solely, can help us improve the performance of TTT?
- What's the meaning of “early'' TTT stage?
- What is effect of (2)? Is this assumption used to explain the reason why using $\mathcal{F}^\prime(x_i)$ to calculate $os_i$ instead of $\mathcal{F}^\prime(x^\prime_i)$?

**References**
- [1] Li, Y., Xu, X., Su, Y., & Jia, K. (2023). On the Robustness of Open-World Test-Time Training: Self-Training with Dynamic Prototype Expansion. 2023 IEEE/CVF International Conference on Computer Vision (ICCV), 11802-11812.

---

### Official Review · Reviewer_brwP · 2024-11-01

**Soundness:** 2
**Presentation:** 2
**Contribution:** 2
**Rating:** 5
**Confidence:** 4

**Summary:**

This paper introduces an improved method for open-world test-time adaptation by integrating contrastive learning through data augmentation, building upon the previously developed Open-World Test-Time Training (OWTTT) approach. The authors identify a limitation in OWTTT, specifically its tendency to misclassify certain weak out-of-distribution (OOD) classes as strong OOD due to insufficient contrastive information. To mitigate this issue, the authors propose Open World Dynamic Contrastive Learning (OWDCL). This method augments input samples with data augmentation and applies additional contrastive learning using the NT-XENT loss function. Extensive experiments show that OWDCL achieves superior performance compared to the original OWTTT method.

**Strengths:**

1. The paper presents a solid motivation for incorporating contrastive learning as an additional regularization strategy within the OWTTT framework, adopting a well-suited method for adaptation.
2. The writing is cohesive, and the proposed approach is explained thoroughly and with clarity.

**Weaknesses:**

1. While the paper is well-executed, its novelty seems quite incremental, with limited unique contributions. Much of the OWDCL framework builds directly on prior work, OWTTT, with NT-XENT serving as an existing method for contrastive learning. Additionally, the performance improvement is also limited, especially when additional augmentations are included.
2. The paper lacks in-depth explanation and empirical evidence to substantiate the claim that "misclassification of certain weak OOD classes as strong OOD results from insufficient contrastive information." Both theoretical support and experimental comparisons are necessary to highlight the effectiveness of contrastive learning in addressing this problem.
3. Contrastive learning for test-time training and adaptation is explored in various existing methods, such as Chen et al. (2022) and Wang et al. (2022). It would be helpful for the authors to provide a comparison to these methods and clarify the distinctions.
4. The current experimental setting may be too simple for the problem. Proof of the proposed solutions should be conducted on more challenging cross-domain datasets such as PACS and DomainNet.


[1] Li, Y., Xu, X., Su, Y., & Jia, K. (2023). On the robustness of open-world test-time training: Self-training with dynamic prototype expansion. *Proceedings of the IEEE/CVF International Conference on Computer Vision*, 11836–11846.

[2] Chen, D., Wang, D., Darrell, T., & Ebrahimi, S. (2022). Contrastive test-time adaptation. *Proceedings of the IEEE/CVF Conference on Computer Vision and Pattern Recognition*, 295–305.

[3] Wang, Q., Fink, O., Van Gool, L., & Dai, D. (2022). Continual test-time domain adaptation. *Proceedings of the IEEE/CVF Conference on Computer Vision and Pattern Recognition*, 7201–7211.

**Questions:**

Please refer to **Weakness**.

---

### Official Review · Reviewer_zyHJ · 2024-11-03

**Soundness:** 2
**Presentation:** 2
**Contribution:** 2
**Rating:** 5
**Confidence:** 4

**Summary:**

This paper proposes to use contrastive learning to address the problem of Open-World Test-Time training. The authors believe that the misclassification of weak OOD data as strong OOD noise is due to the lack of contrastive constraint between the samples. To address this, the authors propose the Open World Dynamic Contrastive Learning (OWDCL) method, which performs contrastive learning between the original samples and the corresponding augmented samples, and within each semantic cluster. To validate the effectiveness of the proposed method, the authors perform experiments on CIFAR10-C, CIFAR100-C and ImageNet-C datasets. The experimental results demonstrate the improvement of the proposed method.

**Strengths:**

1. The paper is well written and easy to follow.
2. Motivation and methodology are clearly stated. The experiments are sufficient.

**Weaknesses:**

1. The novelty is incremental. First, contrastive learning has been shown to be effective in many different tasks. In the TTA task, TTT++, AdaContrast also use contrastive loss in their work. Therefore, the authors need to clarify why contrastive learning used in this work can address OWTTT well and why others cannot achieve it. This analysis can be some theoretical derivations or some targeted contrastive experiments.
2. In Eq. 4 of the manuscript, I am wondering why the L2 norm is performed in the batch channel instead of the feature channel. What is the reason for this?
3. The formulae in the manuscript are unclear and misleading. For example, in Eq.12, the definitions of $k$, $c$ and $K$ are misunderstood. While $k$ is the pseudo label of the test sample and $K$ is the superscript of the sum function in Eq.12, $k$ is redefined as the number of sample pairs and $K$ is the class id in line 270.
4. This paper uses relatively simple data augmentation techniques to generate positive sample pairs. However, the effectiveness of these techniques may be limited when dealing with out-of-distribution (OOD) data that are highly complex or significantly different from the original data, such as different domains in PACS datasets.

**Questions:**

As described in Weakness Section.

---

### Note · Authors · 2024-11-12

I have read and agree with the venue's withdrawal policy on behalf of myself and my co-authors.